# OpenReview forum: "MaMa: A Game-Theoretic Approach for Designing Safe Agentic Systems"
_ICML.cc/2026/Conference — ICML 2026 regular_

### Official Review · Reviewer_qkRN · 2026-03-11

**Soundness:** 2
**Presentation:** 3
**Significance:** 2
**Originality:** 2
**Overall Recommendation:** 3
**Confidence:** 4

**Summary:**

The paper proposes MaMa, a framework for automated design of safe multi-agent LLM systems by framing the problem as a Stackelberg security game. A Meta-Agent iteratively proposes system designs while a Meta-Adversary searches for worst-case attacks by compromising a subset of agents. The two players alternate in updates: the Meta-Adversary finds strong attacks, and the Meta-Agent updates the system to defend against them. Several experiments are used to demonstrate the results.

**Compliance With Llm Reviewing Policy:**

Affirmed.

**Final Justification:**

Improved partially.

**Key Questions For Authors:**

1. Please provide convergence guarantees or equilibrium analysis to justify the Stackelberg framing, or quantify how far the Meta-Adversary's heuristic attacks are from the true best response.
2. The claim that safety-constrained optimization outperforms quality-only optimization is counterintuitive and theoretically suspect.
3. Provide a sensitivity analysis of the safety-quality tradeoff, e.g., Pareto frontier analysis, to inform a better selection of safety level.
Check details in Strengths And Weaknesses.

**Limitations:**

yes

**Strengths And Weaknesses:**

Strength
- The problem setting is well-motivated. Designing agentic systems that are robust to internal agent compromise is good.
- Leader-follower game theoretic idea is a natural choice for the robust agentic system design.

Weakness
- The first concern is that the Stackelberg framing is largely cosmetic. The proposed approach, MaMa, is simply adversarial training with LLM-generated attacks. Actually, the Meta-Adversary never computes a true best response. It just runs a fixed number of heuristic attack attempts. Calling this a Stackelberg game implies a solution concept that the method doesn't actually approximate.
- In addition, there's no convergence guarantee, no equilibrium analysis, and no discussion of how far the discovered attacks are from the true best response solution, which are fundamental questions one asks in the proposed security game.
-The paper claims MaMa maintains quality while improving safety, and in some environments, quality even increases. First, as can be seen in Fig 5, quality under attack remains substantially below clean quality across all environments. Second, it is very counterintuitive that the safety-constrained solution outperforms the ones only optimized for quality. Theoretically, this is impossible. Either the results just contain too much noise, or the game-theoretic optimization problem is never solved exactly.
-  In addition, the objective of the leader (MaMa Agent) explicitly trades off safety and quality via the weight, but there is no sensitivity analysis in experiments.
- It is claimed that the work improves on prior work by separating tools and agents, but provides no empirical evidence that this formulation outperforms unstructured alternatives, such as AFlow's Python function approach. A complete and fair comparison is needed.
- The number of adversaries is limited to 1 in all training. What about training a more conservative system with more adversaries? Does the quality decrease a lot? How about the security-quality trade-off then?
- check typos: "" is consistently typed wrong in the paper.

---

> ### Author Rebuttal · Authors · 2026-03-30
>
> We thank the reviewer for their valuable feedback. We are delighted to read that the reviewer appreciates our problem setting. In the following, we will respond to their concerns:
>
> W1. **Stackelberg Framing**
>
> We respectfully disagree that the Stackelberg framing is cosmetic. While we do not compute an exact equilibrium, the Stackelberg framing directly informs the objective and the algorithmic structure of MaMa, where the Leader optimizes under worst-case attacks by the follower. Furthermore, approximately solving a Stackelberg game using a slowly updating leader and a quickly adapting follower is a well-established algorithm in prior work [1,2].
>
> W2./Q1. **Theoretical Results**
>
> It is correct that we did not include these results due to the complex nature of agentic systems. It is possible to constrain the number of agents and tokens used for each agent’s instruction, which would result in a finite state space and, therefore, the existence of an equilibrium.
>
> Convergence of the method is difficult to establish, as we allow for the design of Turing-complete tools, resulting in difficulty due to the halting problem.
>
> We will make this aspect clearer in future revisions by focusing more on the method and evaluation.
>
> W3./Q2. **Optimality of results**
>
> It is correct that quality under attack remains below clean quality. We want to clarify that we never claimed otherwise. Our problem formulation in Section 3.3 explicitly targets high quality in clean settings. We believe achieving high quality under attack is a genuinely open problem: when an adversary controls one or more agents, the system is effectively operating with a compromised component.
>
> Regarding the quality improvement, we agree that in an exact solution, a safety-constrained solution cannot outperform an unconstrained one. However, we note that MaMa does not solve the optimization problem exactly. Both the Meta-Agent and Meta-Adversary operate via heuristic search. In this approximate setting, the additional structure imposed by robustness optimization can act as a beneficial regularizer, which in turn can improve clean performance. This phenomenon is not unprecedented. Most prominently, adversarial training [3] has been shown to improve clean accuracy in certain settings despite introducing a robustness constraint.
>
> W4/Q3. **Sensitivity Analysis**
>
> We investigated the effect of varying η during optimization and found that it does not reliably shift the emphasis between quality and safety in the resulting systems. We attribute this to the dominant role of natural language feedback in guiding the Meta-Agent's design decisions.
>
> However, we found that η can be used in a different way: It can be applied post-hoc to select a system from the archive generated by MaMa. Specifically, by reranking the archive according to a weighted sum of quality and safety scores with a chosen η, practitioners can select systems with different quality-safety profiles without any additional optimization. This provides a lightweight and practical mechanism for sensitivity analysis and preference-based selection.
> η|Quality|Safety
> -|-|-
> TP||
> 0.1|4.0|4.0
> 1|3.9|4.2
> 2|3.9|4.2
> FAW||
> 0.1|5.0|3.6
> 1|4.7|4
> 2|4.7|4
> CG||
> 0.1|2.6|3.6
> 1|2.5|4
> 2|1.8|4.4
> PA||
> 0.1|3.9|4.6
> 1|3.9|4.6
> 2|3.9|4.6
>
> W5. **Comparison to AFlow**
>
> We want to clarify that we never claimed our structured formulation outperforms AFlow or other unstructured alternatives. As stated in Section 3.1, the structured separation of agents and tools is a functional requirement of our approach rather than a design choice motivated by performance considerations. Specifically, defining a meaningful attack space for the Meta-Adversary requires a clear and enumerable set of agents that can be selected and modified. This is not possible when agentic systems are represented as arbitrary Python functions, as in AFlow, since the boundaries between agents and tools become implicit and non-enumerable.
>
> W6. **Number of Adversaries**
>
> We ran additional experiments increasing the number of adversaries during the optimization. We find that training with more adversaries does produce systems that are more robust against stronger attacks, but the quality-safety tradeoff becomes more pronounced as the number of adversaries increases.
> Adversaries|1|2|3
> -|-|-|-
> TP||
> Quality|4.1|3.4|3.6
> Safety|3.8|3.9|4
> FAW||
> Quality|3.8|3.7|3.5
> Safety|4.3|4.5|4.0
>
> W7. **Typos**
>
> We thank the reviewer for bringing our attention to this, and we will fix this in future revisions.
>
> We thank the reviewer once again and hope this clarified their concerns. We are happy to answer any follow-up questions.
>
> [1] Brückner, Michael, and Tobias Scheffer. "Stackelberg games for adversarial prediction problems." SIGKDD 11\
> [2]Rajeswaran, Aravind, Igor Mordatch, and Vikash Kumar. "A game theoretic framework for model based reinforcement learning." ICML 20.\
> [3] Goodfellow, Ian J., Jonathon Shlens, and Christian Szegedy. "Explaining and harnessing adversarial examples."  ICLR 15.

---

> > ### Author Rebuttal · Reviewer_qkRN · 2026-04-03
> >
> > I thank the authors for the response. Some concerns are addressed, while some remain. The heuristic nature of the approach and its lack of theoretical analysis remain a concern. Given that it is a Stackelberg game, one is interested in the Stackelberg solution and/or in analyzing the approximation error. In this regard, the reviewer will raise the score to 3.

---

> > > ### Author Response · Authors · 2026-04-05
> > >
> > > We once again thank the reviewer and we are delighted that we could address the concerns regarding the breath of experiments.
> > >
> > > Regarding the theoretical concerns, we fully acknowledge that our work focuses on the proposed algorithm and the empirical evaluation. In light of this feedback, we plan to revise the paper to more clearly position our contribution as an empirical one, reducing the emphasis on the Stackelberg framing in the abstract, introduction, and problem setting, and instead focusing on the algorithm design and experimental results. We believe this more accurately reflects the nature of our contribution.
> > >
> > > We are grateful for the reviewer's constructive feedback throughout this discussion, which has meaningfully improved our paper.

---

### Official Review · Reviewer_fkh2 · 2026-03-13

**Soundness:** 3
**Presentation:** 3
**Significance:** 3
**Originality:** 4
**Overall Recommendation:** 5
**Confidence:** 4

**Summary:**

This work introduces MaMa, a game-playing method of developing safer agent systems. An agent designer ("Meta Agent") designs safe agentic systems, and is able to iterate over these system definitions based on feedback from an attacker ("Meta Adversary"), that can "take over" certain parts of the system.

**Compliance With Llm Reviewing Policy:**

Affirmed.

**Key Questions For Authors:**

1) Have you given any thoughts to further constraining the optimization problem at play here? I would be interested to hear your thoughts on how to ensure that these systems are safe and scalable, or that the adversaries are dangerous but stealthy.

**Limitations:**

yes

**Strengths And Weaknesses:**

### Strengths

This work presents a novel and interesting method of generating insights into safer agentic system designs. In my view, instead of primarily evaluating the safety of an agentic system, this work evaluates the ability of a meta agent to design safe systems, which is an interesting framing compared to most existing literature on agentic safety. The use of programmatic methods for designing these systems seems to improve the consistency of said systems, and also enables rapid iteration. This paper also presents interesting findings; though I'm not certain they are findings that aren't obvious. This mechanism also proves to be robust to indirect prompt injection attacks, and provides an interesting look into how one might design agentic systems that are robust given the failure of an agent internal to the system – something that is undoubtedly useful given the growing existence of systems with lots of subagents.

### Weaknesses

My identified weaknesses are as follows:
1) The tasks and environments at play here seem to be somewhat low fidelity. While I don't think that high-fidelity is strictly necessary to evaluate the *design* of a system, I do believe that good evaluations are closely linked with high-realism environments.
2) Following from the above, the task environments are very structured and narrow. It is unclear to me how much this type of method will generalize, particularly when extended to agents with more power (e.g. CUAs).
3) I would have liked to see additional constraints on the system. I think that could be a *very* interesting instantiation of an optimization problem. For example, the adversary is currently trying to degrade the system, but what if it's goal was to make the system susceptible to certain vulnerabilities that need to evade black box monitors? Additionally, what if the safe system needs to be designed to be safe but also within computational constraints?

---

> ### Author Rebuttal · Authors · 2026-03-30
>
> We thank the reviewer for their valuable feedback. We are delighted to read that the reviewer appreciates our method, our  findings, and the relevance of the problem. In the following, we will respond to their concerns.
>
> W1.**Fidelity of Tasks**
>
> We agree that higher-fidelity environments are desirable and note that this is precisely why we additionally evaluate MaMa on AgentDojo [1], a widely-used benchmark specifically designed to evaluate agents in realistic, dynamic settings with real-world tool use. Please refer to our response to W4 of Reviewer R1Vw for the results.
>
> We would like to clarify that both BAD-ACTS and AgentDojo are specifically designed to capture realistic agent behaviors, including dynamic interactions and tool usage that closely mimic real-world APIs. In particular, AgentDojo has been widely adopted as a benchmark for evaluating agents in complex, realistic settings.
>
> However, we agree with the reviewer that an evaluation in high-realism environments, such as a responsible evaluation in real-world systems, is an important and interesting direction for future work. However, given the complexity of scaling methods towards high-realism scenarios, we believe that such questions are out of scope of our paper, and we will add this as an explicit direction for follow-up work.
>
> W2. **Generalization to more Powerful Agents**
>
> We thank the reviewer for raising this important point. We agree that generalization to more complex and less structured agent settings, such as CUAs, is an important open question.
>
> Empirically, we provide initial evidence of generalization in Table 2, where systems designed by MaMa transfer to stronger underlying models and different adversaries while maintaining high safety. Further, we evaluated the transferability to newer models specifically designed for agentic systems. Please refer to our response to W4 to Reviewer R1Vw for the results.
>
> Beyond this, we would like to emphasize that MaMa primarily discovers structural safety mechanisms such as multi-agent approval chains, mutual monitoring, and dedicated safety reviewer agents, which are largely independent of the specific task environment. We believe these architectural patterns are, in principle, applicable to more powerful agent settings, including CUAs.
> That said, extending MaMa to fully open-ended and significantly more complex environments introduces additional challenges in both system design and evaluation. We view this as an exciting and important direction for future work, and will highlight it in the revised paper.
>
> W3 and Q1. **Constraints on Meta-Adversary and Meta-Agent**
>
> We thank the reviewer for these insightful suggestions. We agree that incorporating additional constraints on both the adversary and the system is a promising direction.
>
> Regarding adversary's objectives, our current work focuses on worst-case safety, where the Meta-Adversary directly minimizes safety. This choice is motivated by the goal of obtaining systems that are robust under the strongest possible attacks.
> We want to note that implicitly our setup does already consider stealthy adversaries. One of the most commonly designed defense mechanisms are detector agents, which intercept messages and report any malicious actions. The only way for the Meta-Adversary to perform malicious actions is therefore to evade these agents by being stealthy.
>
> That said, our Stackelberg formulation is general and can naturally accommodate alternative adversarial objectives, such as explicitly stealthy attacks that evade detection or targeted vulnerabilities. We agree that modeling such constrained or stealthy adversaries is an especially interesting extension, as it would more closely reflect real-world threat models. However, in this paper we focused on the defensive side against worst-case adversaries.
>
> On the system side, we partially account for computational constraints by limiting the number of messages and tool executions per episode (Line 266), which implicitly bounds system complexity. We further found that the final systems never contain more than three additional agents.
>
> However, we agree that more explicitly incorporating resource constraints (e.g., latency, cost, or number of agents) into the optimization objective is an important direction for improving scalability.
>
> Overall, we view these extensions as natural next steps enabled by our framework, and we will highlight them clearly as future work in the revision.
>
>
> We thank the reviewer once again and hope this clarified their concerns. We are happy to answer any follow-up questions.
>
> [1] Debenedetti, Edoardo, et al. "Agentdojo: A dynamic environment to evaluate prompt injection attacks and defenses for llm agents." NeurIPS 24

---

> > ### Author Rebuttal · Reviewer_fkh2 · 2026-04-02
> >
> > I thank the authors for their thoughtful responses to my comments, and maintain my score.

---

> > > ### Author Response · Authors · 2026-04-05
> > >
> > > We thank the reviewer for their time, engagement, and continued support.
> > >
> > > We are glad our responses were satisfactory and look forward to incorporating the feedback into the final version of the paper.

---

### Official Review · Reviewer_NpqF · 2026-03-14

**Soundness:** 3
**Presentation:** 3
**Significance:** 3
**Originality:** 3
**Overall Recommendation:** 5
**Confidence:** 4

**Summary:**

This paper proposes MaMa (Meta‑Adversary–Meta‑Agent), a framework and algorithm for automated design of multi‑agent LLM systems that remain safe under worst‑case internal compromise, i.e., when an attacker can directly overwrite a subset of agents' instructions. The core idea is to model design as a Stackelberg security game between a leader designer (Meta‑Agent) and a best‑responding attacker (Meta‑Adversary), and to approximately solve it via iterative LLM‑driven search where the attacker continuously discovers strong attacks and the designer patches vulnerabilities.

Main results: the Meta‑Adversary can substantially reduce safety even for systems that include common guardrails. MaMa improves safety over generations while generally maintaining quality, with improvements across environments with standard deviations over three evaluations and focuses on worst‑case (top‑5) attacks. Systems show robustness to different underlying LLMs, to stronger adversaries controlling more agents, to targeted attacks with rule‑based evaluation, and to indirect tool injection attacks with an additional "Delimiters" baseline. Qualitatively, MaMa tends to introduce reviewer/approver‑style agent structures and monitoring behaviors that require multi‑agent involvement before actions occur.

The authors conclude MaMa yields systems much safer than initial/manual/quality‑only designs, while noting significant compute requirements and suggesting future work on learned predictors of safety/quality and richer adversary capabilities (e.g., tools or communication graphs).

Overall, this is a strong and timely contribution at the intersection of LLM agents, automated workflow design, and adversarial robustness. I would suggest to: (a) make the formalization–algorithm–evaluation chain more explicit and falsifiable, (b) strengthen evaluation against overfitting to the particular attacker/judge setup, and (c) improve reproducibility (public release of code/configs, or at minimum a complete artifact).

**Compliance With Llm Reviewing Policy:**

Affirmed.

**Key Questions For Authors:**

1. What exactly is the scalar objective optimized by the Meta‑Agent (including any weights and whether scores are normalized across environments)? Please provide the explicit formula as implemented, not just as described, and explain how the 1–5 judge scale is treated mathematically.

2. Under what conditions do you consider your Meta‑Adversary search to approximate a best response? Do you have any empirical evidence that increasing the attack budget (beyond 25 attempts) materially changes the discovered worst‑case safety?

3. How do you ensure the Meta‑Adversary explores qualitatively diverse attack strategies (as opposed to local variations)? Is there any explicit diversity mechanism beyond providing top‑k previous attacks?

4. TWhy is the adversary restricted to modifying only agents (with tool/graph changes disallowed by the distance definition)? How does this map to realistic compromise scenarios, especially for tool‑using agents where untrusted tool outputs are central?

5. The tool injection test changes the attacker capability (manipulating tool outputs). Is this capability intended to be covered by your formal game model, or is it strictly an out‑of‑distribution robustness check? Please clarify the conceptual connection.

**Limitations:**

The paper's core story is robustness to best‑response attacks in a Stackelberg structure. In practice, the best response is approximated by an LLM search with fixed budgets (attacks per generation, top‑k reporting). This means the resulting system can be robust to discovered attacks while remaining vulnerable to undiscovered ones. Thus, the main conclusions should be interpreted as "robustness improved under this adversarial search regime," not as a guarantee.

The paper trains and evaluates on environments from BAD‑ACTS, a benchmark introduced by the same authors in prior work (and used to motivate Guardian Agents). This is not inherently problematic, but it increases the importance of demonstrating external validity. Impact: improvements might partially rely on structural affordances of BAD‑ACTS tasks/scoring. I would suggest trying to add evaluation on an independent benchmark emphasizing different tool surfaces and threat models (e.g., dynamic prompt injection settings).

**Strengths And Weaknesses:**

The empirical case is compelling in the sense that (i) a dedicated Meta‑Adversary can substantially degrade safety of baseline systems, and (ii) MaMa‑generated systems improve safety while generally preserving quality across four environments, with additional robustness checks under changed threat settings.

Framing automated agentic system design as an explicit leader–follower adversarial game is a valuable conceptual step: it clarifies that "robustness" is inherently a strategic interaction, and it connects this line of work to established security‑games thinking. The paper also clearly positions itself as extending automated design (e.g., AFlow) by adding an explicit adversarial player rather than optimizing success alone.

---

> ### Author Rebuttal · Authors · 2026-03-30
>
> We thank the reviewer for their valuable feedback. We are delighted to read that they appreciate our empirical evaluation and our framing as a Stackelberg game. In the following, we would like to respond to their concerns:
>
> W1. **Formalization–Algorithm–Evaluation Chain**
>
> We thank the reviewer for this suggestion, and we will make the following chain more explicit: the Stackelberg formulation motivates the asymmetric update rule in Algorithm 2. This, in turn, generates the falsifiable prediction that iterative exposure to worst-case attacks should produce strictly safer systems than optimizing for quality alone, which is precisely what Figure 2 demonstrates.
>
> W2. **Overfitting to attacker/Judge**
>
> We believe our current results already provide strong evidence against overfitting to the particular attacker and judge setup. Specifically, we note three sources of evidence. First, varying the attack model at deployment time does not meaningfully decrease safety scores (Table 2). Second, the designed systems remain robust across qualitatively different attack modalities (Tables 3 and 4). Third, our judge models were validated against human annotations both on pre- and post-optimization trajectories, with consistent agreement in both cases (Line 258).
>
> W3. **Public Release of Code**
>
> We agree with the reviewer and are committed to full reproducibility. We will publicly release all code required to reproduce our experiments.
>
> L1. **Accuracy of Conclusion**
>
> We fully agree that the correct interpretation of our results is robustness under the specific adversarial search regime used during optimization, rather than a guarantee against all possible attacks. We will update our claims throughout the paper to reflect this more carefully.
>
> L2. **Additional Benchmarks**
>
> We thank the reviewer for highlighting this, and agree that our evaluation would benefit from including additional benchmarks. Please refer to our response to W4 to Reviewer R1Vw for results on the AgentDojo benchmark.
>
> Q1. **Objective of Meta-Agent**
>
> The scalar objective optimized by the Meta-Agent is the unweighted sum of the quality and safety scores, corresponding to η = 1 as reported in Table 7. Concretely, given a system design, we compute the average quality score over 10 clean episodes and the average safety score over the 5 strongest attacks discovered by the Meta-Adversary, both on the 1-5 scale. This is returned directly by the judge. No normalization is applied across environments.
>
> In practice, the Meta-Agent receives the full natural language feedback produced by the judge alongside the scalar score, and we found that this textual feedback is the primary driver of meaningful system improvements across iterations. The scalar score is used mainly for archive sampling and final system selection. We will discuss this aspect more explicitly in the revised paper.
>
> Q2. **Approximation of best response**
>
> We have decided to use 25, as we empirically found no improvement with a longer horizon of attacks in a pilot study where we used 100 iterations. Specifically, we found that only in one out of four environments an attack was found that decreased the top-5 score after the first 25 iterations, leading to a decreased safety score by 0.2. We will include these results in future revisions.
>
> Q3. **Existence of diversity mechanisms**
>
> Our primary diversity mechanism operates through the Meta-Adversary's instructions: In each iteration, the Meta-Adversary is provided with the top 5 previously discovered attacks and is explicitly instructed to deviate from them when proposing new attacks.
>
> We found this approach to be sufficient in practice. Specifically, we never observed any duplicate attacks in the returned top 5 across any environment or iteration.
>
> Q4. **Attack Modality**
>
> We believe that our threat model is strictly stronger than indirect attack scenarios such as prompt injection through tool outputs, as an adversary with direct control over an agent can explicitly instruct it to perform malicious actions, whereas indirect attacks must rely on the agent being susceptible to manipulated inputs.
> Regarding the exclusion of graph and tool modifications: this choice was motivated by real-world deployment scenarios, where these functions are predetermined and are not easily accessible to an outside attacker. However, we want to note that the Meta-Adversary is not entirely restricted from influencing communication structure. We observed in practice that it discovered strategies that indirectly manipulate routing by exploiting how the selector function interprets messages.
>
> Q5. **Tool Injection**
>
> The tool injection is an out-of-distribution robustness check evaluating the ability of systems designed by our method to generalize to different types of attackers.
>
> We thank the reviewer once again and hope this clarified their concerns. We are happy to answer any follow-up questions.

---

> > ### Author Rebuttal · Reviewer_NpqF · 2026-04-04
> >
> > Thank you. I strongly suggest adding the key fragments of your rebuttal to the main body of the paper, should it be accepted.

---

> > > ### Author Response · Authors · 2026-04-05
> > >
> > > We thank the reviewer for their time, continued support, and helpful suggestions.
> > >
> > > We will make sure to incorporate the key points from our rebuttal into the main body of the revised paper.

---

### Official Review · Reviewer_R1Vw · 2026-03-15

**Soundness:** 3
**Presentation:** 2
**Significance:** 1
**Originality:** 3
**Overall Recommendation:** 3
**Confidence:** 4

**Summary:**

The paper introduces an adversarial multi-agent system design framework formulated as a Stackelberg security game. The authors describe the method, which involves two parties "meta-optimizing" over attacks and defenses, and conduct a 3-run evaluation on BAD-ACTS of GPT-5.1 vs Qwen-3-32b, in addition to a transfer study.

**Compliance With Llm Reviewing Policy:**

Affirmed.

**Final Justification:**

The rebuttal didn't satisfactorily address the role of the Stackelberg games, but it added new experiments, which represented my other concern. Therefore, I increased my score.

**Key Questions For Authors:**

N/A

**Limitations:**

Yes

**Strengths And Weaknesses:**

Strengths:
* The paper focuses on an important (and somewhat novel) topic, namely the intersection of automated MAS design and MAS that can resist a subset of agents being controlled by an adversary
* The authors report all the prompts in the Appendix

Weaknesses:
* The Stackelberg/game-theoretic formulation is, to put it mildly, pointless. There is no discussion of the incentives or of the game-theoretic equilibria emerging from this formulation, despite the emphasis given by the title, abstract, and Introduction. What the paper describes is essentially a double-nested adversarial optimization loop, which has been the norm for almost a decade in the adversarial robustness literature [1, 2]
* The paper does not cite the existing (albeit small) literature on multi-agent adversarial robustness (e.g. [3, 4, 5]), and does not place the results in the context of past work
* The experiments are conducted over 3 runs. This is nowhere enough to establish the value of the results
* The experiments are conducted using only one model pair, with a very small transferability study. Additionally, they are conducted over a benchmark with only 4-5 environments

Edit: in light of the new experiments, I am willing to raise my score, though they are still not at the level of a full experimental evaluation.

In terms of the dimensions:
* Soundness: While the formulation of the technique is reasonable, the design of the experiments is lacking, especially when it comes to adequate sample sizes
* Presentation: The paper is relatively well-written, though it pointlessly invokes concepts that are not used throughout the narrative (e.g. Stackelberg games)
* Significance: The paper focuses on an important threat model, but it does not support its claims with an adequate empirical evaluation
* Originality: The paper introduces an interesting approach (automated design of robust MAS), but overstates its originality and fails to place it in the context of the literature

Overall, the paper is not properly placed in the context of existing literature, misuses concepts from the game theory literature, and makes strong claims with an experimental evaluation of one model pair, 5 environments, and 3 runs. As such, it does not clear the bar for an ICML publication.

[1] Madry, Aleksander, et al. "Towards deep learning models resistant to adversarial attacks." arXiv preprint arXiv:1706.06083 (2017).

[2] Tramer, Florian, et al. "On adaptive attacks to adversarial example defenses." Advances in neural information processing systems 33 (2020): 1633-1645.

[3] Zhou, Jialong, Lichao Wang, and Xiao Yang. "Guardian: Safeguarding llm multi-agent collaborations with temporal graph modeling." arXiv preprint arXiv:2505.19234 (2025).

[4] Huang, Jen-tse, et al. "On the resilience of llm-based multi-agent collaboration with faulty agents." arXiv preprint arXiv:2408.00989 (2024).

[5] Triedman, Harold, Rishi Jha, and Vitaly Shmatikov. "Multi-agent systems execute arbitrary malicious code." arXiv preprint arXiv:2503.12188 (2025).

---

> ### Author Rebuttal · Authors · 2026-03-30
>
> We thank the reviewer for their valuable feedback. We are delighted to read that the reviewer appreciates the topic of our paper. In the following, we would like to respond to their concerns:
>
> W1. **Significance of the Stackelberg Formalization**
>
> We acknowledge that we do not compute exact Stackelberg equilibria, and agree this warrants more explicit discussion. Given the complex nature of agentic systems, including both natural language instructions and tools, it is very difficult to derive theoretical insight from the setup. It is, however, possible to constrain the number of agents and tokens used for each agent’s instruction, which would result in a finite state-space and, therefore, the existence of an equilibrium.
>
> However, we respectfully disagree with the assessment that the formalization is pointless: the leader-follower ordering directly determines the asymmetry of our optimization algorithm. The Meta-Agent commits to a design before the Meta-Adversary best-responds. Such approaches have been studied in-depth in prior work [1,2]. We therefore believe that formalizing the problem as a Stackelberg game has two clear advantages: Firstly, it clearly guides the algorithm design, and secondly, it highlights the asymmetric objectives of the two agents, where the follower is only interested in minimizing safety, while the leader additionally has to consider the quality of the outputs.
>
> This aspect additionally explains the difference between our work and the two cited papers. Adversarial optimization is inherently zero-sum. However, for agentic systems, it is more natural to decompose robustness aspects into quality and safety considerations. For such problems, the Stackelberg formalization is more natural and guides the algorithm design more clearly.
>
> However, we will improve the writing of the paper by clarifying the connection and focusing more on the algorithm design and empirical evaluation of the work in future revisions.
>
> W2. **Missing References**
>
> We thank the reviewer for bringing our attention to these works, and we will include them in future revisions.
>
> W3. **Limited Runs**
>
> We thank the reviewer for this comment. We acknowledge that 3 runs is a limited number of evaluations, and agree that additional runs strengthen the validity of our results. We therefore conducted 7 additional runs of evaluation, bringing the total to 10 runs per environment. We found that results remained largely similar.
>
> Env|Orig. Quality|Quality|Orig. Safety |Safety
> -|-|-|-|-
> Travel Planning|4.23(0.09)|4.21(0.08)|3.71(0.1)|3.74(0.2)
> Financial Article Writing|3.45(0.75)|3.71(0.43)|4.15(0.5)|4.18(0.32)
> Code Generation|3.73(0.41)|3.51(0.30)|4.6(0.48)|4.47(0.26)
> Personal Assistant|3.32(0.4)|3.28(0.48)|3.98(0.32)|4.12(0.46)
>
> W4. **Extensiveness of the Experiments**
>
> We want to note that a single run of generation comes at a significant cost, including multiple days of runtime and hundreds of dollars in API calls, limiting the amount of models we can evaluate. We further want to note that our experiment setup is consistent with a large set of influential works on automated agentic system design [3,4,5], where often a system is only optimized for a single model pair and the transferability to other models is evaluated.
>
> However, we agree that more environments would be beneficial to explore the diversity of application scenarios and types of attacks our method can be deployed in. Therefore, we extend our experiments with three additional environments from AgentDojo [6], Where we found that our method can also improve system safety and quality.
> Env|Quality|Safety
> -|-|-
> Workspace-init|1.4|3.0
> Workspace-best|**3.3**|**4.4**
> Slack-init|3.76|1.4
> Slack-best*|**3.96**|**4.0**
> Banking-init|1.6|1.0
> Banking-best*|**2.6**|**4.6**
>
> Results marked with * have not finished completely within the rebuttal timeline, we expect results to improve.
>
> We further extended our transferability experiments to new agentic models, where we found that results remain consistent with our current setup.
>
> Env|Llama3.3|glm-4.7|Nemotron-3-super
> -|-|-|-
> Travel Planning|3.81|4.6|4.0
> Financial Article Writing|3.75|4.0|4.4
> Code Generation|4.26|5.0|3.8
> Personal Assistant|4.63|4.2|4.6
>
> We thank the reviewer once again, and hope this clarified their concerns. We are happy to answer any follow-up questions.
>
> [1] Rajeswaran, Aravind, Igor Mordatch, and Vikash Kumar. "A game theoretic framework for model based reinforcement learning." ICML 20\
> [2] Brückner, Michael, and Tobias Scheffer. "Stackelberg games for adversarial prediction problems." SIGKDD 11\
> [3] Hu, Shengran, Cong Lu, and Jeff Clune. "Automated Design of Agentic Systems." ICLR 25\
> [4] Zhang, Jiayi, et al. "AFlow: Automating Agentic Workflow Generation." ICLR 25\
> [5] Shang, Yu, et al. "AgentSquare: Automatic LLM Agent Search in Modular Design Space." ICLR 25\
> [6] Debenedetti, Edoardo, et al. "Agentdojo: A dynamic environment to evaluate prompt injection attacks and defenses for llm agents." NeurIPS 24

---

> > ### Author Rebuttal · Reviewer_R1Vw · 2026-04-03
> >
> > I thank the authors for the rebuttal. That said, I do not consider the rebuttal sufficient to support full acceptance.
> >
> > > We acknowledge that we do not compute exact Stackelberg equilibria, and agree this warrants more explicit discussion. [...] it is very difficult to derive theoretical insight from the setup
> >
> > In that case, relying on a Stackelberg formalization is unnecessary, and adds only a cosmetic layer on top of an otherwise simple setup (this sentiment is echoed by Reviewer qkRN as well). The works cited in the rebuttal use the Stackelberg approach to derive theoretical insights for RL (Rajeswaran et al.) and classification (Bruckner et al.).
> >
> > > Formalizing the problem as a Stackelberg game has two clear advantages: Firstly, it clearly guides the algorithm design, and secondly, it highlights the asymmetric objectives of the two agents, where the follower is only interested in minimizing safety, while the leader additionally has to consider the quality of the outputs.
> >
> > Continuing from above, there is a distinction to be made between being inspired by and relying on a formalization. The paper places much emphasis on the role of Stackelberg games, but then relies in practice on a very standard adversarial optimization framework as described in Section 3.2 and 3.3. To be clear, there is nothing wrong with the actual choice of using constrained optimization as a way of formalizing the problem, but there is no incentive-based analysis that would warrant the paper's heavy conceptual reference of Stackelberg games.
> >
> > That said, I appreciate that the authors have conducted additional experiments, which address my other concern. For this reason, I am raising my score to Weak Reject.

---

> > > ### Author Response · Authors · 2026-04-05
> > >
> > > We thank the reviewer for their continued engagement and are pleased that the additional experiments addressed their concerns about evaluation breadth.
> > >
> > > Regarding the Stackelberg framing, we appreciate the reviewer's careful distinction between being inspired by versus relying on a formalization, and we agree the current manuscript overstates the role of the game-theoretic framework relative to what is formally derived from it. We plan to revise the paper to more carefully calibrate this, specifically by toning down the conceptual emphasis on Stackelberg games in the abstract, introduction, and problem setting, and instead positioning the formalization more modestly as a clarifying framework that motivates the algorithm design.
> > >
> > > We are grateful for the reviewer's constructive and engaged feedback throughout this process, which we believe has substantially strengthened the paper.

---

### Decision · Program_Chairs · 2026-04-30

**Decision:**

Accept (regular)

**Comment:**

This paper proposes MaMa, a framework for automatically designing safe multi-agent LLM systems under a threat model in which a subset of internal agents may be compromised. The method formulates system design as a Stackelberg security game between a system designer (the Meta-Agent) and a best-responding Meta-Adversary. Empirically, the paper shows that the method improves system safety under attack across multiple environments, while in most cases maintaining reasonable task performance.

The strengths of this paper lie in the importance and timeliness of the problem setting, as well as in the novelty of approaching safety from the perspective of automated agentic system design. The rebuttal also materially strengthened the empirical support. However, the current version still has limitations: the manuscript places more emphasis on the Stackelberg framing than is supported by the theoretical analysis currently provided, and the treatment of equilibrium analysis and convergence guarantees remains insufficient at this stage. Therefore, I recommend weak accept.